# High-Fidelity 3D Scene Representation via HDR-Integrated Multi-Constraint Neural Rendering

## Abstract

Recent advances in neural rendering have markedly improved 3D reconstruction and novel view synthesis, yet methods still degrade under complex illumination, weak or low-texture regions, and cross-view inconsistencies from camera ISP pipelines. We propose a unified scene representation framework that densifies geometric supervision via depth-guided virtual view generation plus multi-view consistency priors, improving fidelity in weak-texture and noisy areas; enforces view-independent radiance consistency through bilateral filtering that removes ISP enhancement residuals, decoupling in-camera processing from radiance field optimization; and performs semantics-guided deferred 3D Gaussian field reconstruction fusing pretrained high-level semantic features with material parameters for challenging materials and lighting. It further models scene radiance explicitly with a learnable asymmetric tone-mapping grid to more accurately infer pixel colors and maintain HDR detail, and employs a coarse-to-fine optimization schedule improving stability and convergence. Experiments across indoor and outdoor datasets show consistent quantitative and qualitative gains in reconstruction fidelity and novel view synthesis, with robustness under sparse inputs, weak textures, complex illumination, and HDR conditions, underscoring the benefits of integrating geometric, photometric, and semantic priors in real-world deployments.

## 1 Introduction

3D scene representation, as a core area of computer vision and graphics (Yan et al., 2024; Wu et al., 2024; Mumuni & Mumuni, 2022), is undergoing a transition from traditional modeling to complex, semantically rich representations (Caesar et al., 2020). Recently, Neural Radiance Fields (NeRF) (Mildenhall et al., 2021) and its derivatives have achieved remarkable success in high-quality novel view synthesis (Barron et al., 2022; Fridovich-Keil et al., 2023). However, their reliance on large-scale high-quality training data and high computational costs limits their practical applications (M"uller et al., 2022; Niemeyer et al., 2022). 3D Gaussian Splatting (3DGS) (Kerbl et al., 2023; Wang et al., 2025; Wu et al., 2024)significantly improves training and rendering speed through explicit representation and rasterization techniques (Fridovich-Keil et al., 2023; Chen et al., 2022). Nevertheless, it suffers from limitations in geometric precision and handling complex lighting and non-Lambertian surfaces (Verbin et al., 2022; Yao et al., 2022; Huang et al., 2022).

To address shortcomings, this paper proposes a framework. First, we focus on enhancing reconstruction by introducing multi-view geometric consistency priors, which improve reconstruction in sparse and weak-textured regions (Niemeyer et al., 2022; Chen et al., 2021). Second, to mitigate the interference of camera image signal processing (ISP) on multi-view consistency, we design a view synthesis that decouples these effects, ensuring realistic rendering (Mildenhall et al., 2023; Huang et al., 2022). Lastly, to tackle the challenges of high gloss and complex lighting scenarios, we incorporate semantic guidance and deferred rendering techniques (Kerr et al., 2023; Zhi et al., 2021). Specifically, we redefine the color representation of Gaussian points as radiance and employ a learnable asymmetric grid to model tone mapping from radiance to LDR pixel values, enabling accurate HDR scene reconstruction. Additionally, we adopt a coarse-to-fine optimization strategy to enhance robustness and convergence speed of the model in complex scenarios (M"uller et al., 2022; Chen et al., 2022).

Figure 1: The proposed neural rendering framework integrates multi-view constraints to achieve high-fidelity 3D scene representation. It consists of three key modules: geometric reconstruction guided by consistency priors for weak-texture regions, bilateral-filter-based consistency constraints to optimize radiance fields, and radiance representation with asymmetric tone mapping for HDR material modeling. Together, these modules enable efficient and robust rendering of complex scenes.

Our contributions are summarized as follows:

1. We propose a geometric reconstruction algorithm that introduces multi-view geometric consistency priors. (Chen et al., 2021; Niemeyer et al., 2022) By leveraging depth-guided virtual view generation and feature-level regularization loss, our approach significantly improves the geometric accuracy and robustness of 3DGS in sparse data and weak-textured scenes. (Kerbl et al., 2023)

2. We design a view synthesis algorithm based on bilateral filtering and view-independent consistency constraints. By decoupling camera ISP effects on multi-view consistency and combining global appearance embeddings with pixel-wise bilateral filtering (Kerbl et al., 2023), we propose a 3D Gaussian field reconstruction algorithm guided by large-model semantics and deferred rendering techniques. Utilizing high-level semantic information and implicit material dictionaries, our approach enhances modeling capabilities for complex materials and environmental lighting, enabling the transformation of real-world objects into high-quality digital assets. (Tomasi & Manduchi, 1998; Radford et al., 2021; Thies et al., 2019)

3. We introduce radiance as the color representation for Gaussian points and employ a learnable asymmetric grid to achieve tone mapping from radiance to LDR colors. (Debevec & Malik, 1997; Reinhard et al., 2002) This effectively captures high dynamic range information and improves HDR scene reconstruction accuracy. (Huang et al., 2022)

4. We adopt a coarse-to-fine optimization strategy, initializing the tone mapping function with a simplified monotonic function during the early training phase and introducing a more complex grid-based tone mapper for joint optimization in later stages. (Barron et al., 2022; M"uller et al., 2022) This significantly accelerates convergence and enhances the model's robustness under extreme exposure conditions. (Huang et al., 2022)

## 2 RELATED WORKS

**Camera Model** The ideal pinhole camera model serves as the foundation in computer vision for describing how 3D points are projected onto 2D images(Hartley & Zisserman, 2004). The projection process involves the intrinsic matrix $K$ and the extrinsic matrix $[R \mid t]$, which map a point $P = (X, Y, Z)$ in the world coordinate system to a point $p = (u, v)$ on the image plane:(Zhang, 2000)

$$\begin{bmatrix} u \\ v \\ 1 \end{bmatrix} = K \cdot \frac{1}{Z_c} \begin{bmatrix} X_c \\ Y_c \\ Z_c \end{bmatrix}$$

where $(X_c, Y_c, Z_c)$ represents the coordinates of the point in the camera coordinate system.

**Graphics Rendering Pipeline** Rendering is the process of converting 3D physical representations into 2D images(Foley et al., 1996). Rasterization is a key step, mapping geometric primitives (e.g., triangles) onto screen pixels. Surface shading calculates the final color of each pixel based on light sources, viewing angles, and material properties, with common models including the Lambertian diffuse reflection model and the Phong shading model(Phong, 1975). Texture mapping adds details to object surfaces by mapping 2D images onto 3D surfaces using texture coordinate functions.

**Multi-View Stereo Vision** Multi-view stereo vision (MVS) aims to reconstruct the 3D structure of a scene from images captured from multiple viewpoints(Seitz et al., 2006). Epipolar constraints form the basis of geometric relationships between stereo views (Hartley & Zisserman, 2004). Stereo matching identifies corresponding point pairs across different views to compute depth information, with normalized cross-correlation (NCC) commonly used to measure similarity(Scharstein & Szeliski, 2002). Triangulation calculates the 3D coordinates of points using known camera parameters and the projection positions of matched points.

**Geometry Reconstruction Based on 3D Gaussian Splatting** 3D Gaussian Splatting (3DGS) (Kerbl et al., 2023; Chen et al., 2024; Hess et al., 2025; Tonderski et al., 2024) is a novel technique for 3D reconstruction and rendering that represents 3D scenes using learnable Gaussian ellipsoids. Each Gaussian is defined by its position, covariance matrix (parameterized by rotation and scaling), transparency, and color attributes. Its explicit representation supports dynamic scene modeling and physical simulation.

## 3 METHODOLOGY

In this section, we present the methodology behind our proposed framework for addressing the geometric accuracy limitations of 3D Gaussian Splatting (3DGS) in complex scenes. We start by discussing the techniques used to enforce multi-view geometric consistency and view-independent consistency constraints, which improve the robustness of 3D reconstruction under sparse data and weak texture regions. Next, we introduce our semantic-guided modeling approach for complex illumination and material properties, leveraging high-level features and deferred rendering techniques. Finally, we describe our radiance representation and asymmetric tone mapping strategy, which enables high-fidelity HDR scene reconstruction and rendering.

### 3.1 OVERALL ALGORITHM DESIGN

The goal of this work is to achieve efficient and high-fidelity HDR scene geometry reconstruction and rendering from multi-view LDR RGB images of static scenes. The proposed approach begins by transforming the 3D Gaussian model into an approximate 2D planar representation. Subsequently, multi-view geometric consistency, view-independent consistency, and semantic guidance are utilized to optimize the geometry and appearance of the Gaussian field. Furthermore, radiance representation for Gaussian points and a learnable asymmetric tone mapper are introduced to accurately capture HDR information. A coarse-to-fine optimization strategy is adopted to enhance robustness and convergence speed in complex scenarios. The overall algorithm design is illustrated in Figure 1.

### 3.2 INCORPORATING MULTI-VIEW GEOMETRIC CONSISTENCY PRIORS

To address the geometric accuracy limitations of 3DGS, geometric constraints are introduced to ensure that the 3D Gaussian field closely conforms to real surfaces, especially in sparse data and weak texture regions. The schematic diagram of the geometric consistency constraint algorithm is shown in Figure 2.

**Flattened Feature Representation and Unbiased Depth Rendering** The 3D Gaussian model is compressed into 2D planar Gaussians for more accurate representation of real scene surfaces. Each Gaussian is minimized along its smallest scaling factor direction, and view directions are used to resolve normal ambiguities. The plane-to-camera-center distance and normal maps are rendered and converted into depth maps. This depth rendering method achieves higher geometric consistency and accuracy compared to traditional methods and is unaffected by cumulative weights.

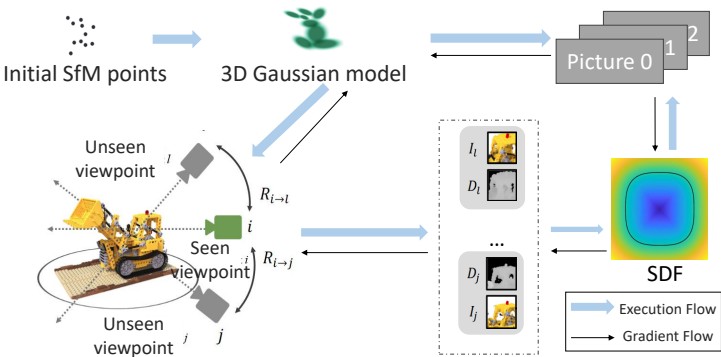

Figure 2: Multi-view geometric consistency pipeline: initial SfM points → flattened planar Gaussians → unbiased depth and normal rendering; single-view ($L_{svgeo}$) and occlusion-aware multi-view geometric and photometric losses ($L_{geo}$, $L_{ap}$) refine the 3D Gaussian field toward accurate surfaces in sparse and weak-texture regions.

**Single-view Consistency Regularization** Under the local planarity assumption, normals are computed from the rendered depth map, and their differences with the rendered normal map are minimized to ensure local depth and normal consistency. Weights are reduced at image edges to achieve smoother geometry. The loss function is defined as:

$$L_{svgeo} = \frac{1}{W_1} \sum_{p \in W} \|\nabla I\|_S \|N_d(p) - N(p)\|_1$$

**Multi-view Consistency Regularization** Due to the discrete nature of Gaussian point cloud optimization, single-view consistency alone is insufficient to ensure global geometric consistency. Multi-view geometric consistency regularization is introduced by detecting occlusions and geometric errors through forward and backward projection errors, excluding occluded pixels. Additionally, photometric multi-view consistency constraints based on planar blocks are adopted, using normalized cross-correlation (NCC) to measure the consistency of diffuse color maps. The final multi-view consistency loss is defined as:

$$L_{mvs} = L_{geo} + L_{ap}$$

### 3.3 VIEW-INDEPENDENT CONSISTENCY WITH BILATERAL FILTERING

To mitigate the impact of modern camera ISP on multi-view consistency, this paper introduces an ISP decoupling strategy during 3DGS training to reduce its influence on scene reconstruction and novel view rendering.

**Image-specific embeddings** are introduced to model scene appearance under varying capture conditions, and affine mappings are predicted via MLP to adjust Gaussian colors:

$$c' = Ac + b$$

**Uncertainty modeling** is employed to identify and ignore transient objects and occluded regions in training images using DINOv2 features and robust loss functions.

**Bilateral grids** are introduced as learnable proxies for ISP, efficiently approximating nonlinear, local, and edge-aware photographic transformations. A bilateral grid $B \in \mathbb{R}^{B_x \times B_y \times B_z \times 12}$ stores affine color transformation coefficients. During training, a bilateral grid is assigned to each training view to decouple camera pipeline-induced differences. A total variation (TV) term is added to regularize the smoothness of the bilateral grid.

### 3.4 COMPLEX ILLUMINATION AND MATERIAL MODELING GUIDED BY SEMANTIC FEATURES

To address the limitations of 3DGS in handling non-Lambertian surfaces and specular reflections, this paper proposes a 3D Gaussian field reconstruction algorithm that combines high-level semantic

information with deferred rendering techniques. For large-scale unbounded outdoor scenes, a feature encoding network based on point cloud environmental bounding spheres is proposed to model sky and ambient lighting information. Sparse sampling and Fourier feature encoding are used to encode foreground and infinitely distant environmental information into high-dimensional features.

**Semantic Feature Distillation and Implicit Material Dictionary**   Multi-level feature maps are extracted using convolutional networks as semantic features $F_s$, guided by pre-trained models such as LSeg. To improve efficiency, an encoder-decoder is employed to compress the semantic feature dimensions. Additionally, an implicit material dictionary $\theta_{material}$ is constructed to map semantic features and Gaussian point positions to material properties (diffuse color $d$, reflectance $r$, residual component $\Delta c$):

$$d, r, \Delta c = \theta_{material}(f, p)$$

**Deferred Shading in Gaussian Splats**   The physical rendering equation is adopted, decomposing BRDF into diffuse and specular components. During rendering, screen-space diffuse color maps, surface normal maps, and reflectance maps are generated first. Subsequently, specular reflection colors are calculated using these maps and learnable environment maps via deferred rendering techniques. The final pixel color is a combination of diffuse and specular colors:

$$C_d(v) = (1 - R(v))C(v) + R(v)E\left(\frac{\|N(v)\|_2 v \cdot N(v)}{N(v)} - v\right)$$

### 3.5 Radiance Representation and Asymmetric Tone Mapping

Inspired by HDRGS, the core rendering process of 3DGS is improved to better support HDR scene reconstruction.

**Radiance Representation**   Traditional 3DGS represents Gaussian point colors directly as RGB values. To accurately simulate the physical imaging process and capture HDR scene information, Gaussian point colors are redefined as radiance $L$. When rasterized onto the image plane, Gaussian points no longer directly represent color $C$ but pixel radiance $E$:

$$E(p) = \sum_{i=0}^{N} L_i \alpha_i \prod_{j=1}^{i-1}(1 - \alpha_j)$$

where $E(p) \in (0, +\infty)$. This enables the model to reconstruct HDR radiance fields, aligning better with real-world physical conditions.

**Learnable Asymmetric Tone Mapper**   Pixel radiance $E$ received by sensors accumulates over exposure time $t$ and is converted to LDR pixel values $C$ via the camera response function (CRF) $F(\cdot)$:

$$C(p, t) = F(E(p) \cdot t(p))$$

Following Debevec and Malik, $F(\cdot)$ is monotonic and invertible. This paper simplifies $F(\cdot)$ as:

$$C(p, t) = g(\ln E(p) + \ln t(p))$$

where $g = (\ln F^{-1})^{-1}$ is the tone mapping function. A learnable asymmetric grid $g_{leaky}$ is designed to model this tone mapping function, enabling better handling of uneven radiance distributions. Dense nodes are used in radiance-dense regions, while sparse nodes are used in radiance-sparse regions, providing greater flexibility and expressiveness.

**Coarse-to-Fine Optimization Strategy**   Directly using grids as tone mapping functions and jointly training them with Gaussian point attributes can lead to severe coupling and overfitting. To address this, a coarse-to-fine optimization strategy is designed:

- **Coarse Phase**: A fixed, monotonic smoothing function (e.g., Sigmoid) is used as the tone mapper, and only Gaussian point attributes are trained. This stabilizes model initialization and avoids local optima.

Figure 3: Radiance and tone-mapping pipeline: 3D Gaussian rasterization accumulates HDR radiance $E$; exposure-adjusted log-radiance is first stabilized by a fixed monotonic (coarse) mapper then refined via a learnable asymmetric grid tone mapper $g_{leaky}$ (multi-resolution bilateral grid with trilinear interpolation, affine transform/matrix and decoder) to produce the updated LDR image.

- **Fine Phase**: The previously designed asymmetric grid is used as the tone mapper, and its parameters are jointly trained with Gaussian point attributes. With good initialization from the coarse phase, the model can effectively learn complex tone mapping relationships and improve reconstruction quality.

## 3.6 LOSS FUNCTIONS

In addition to pixel-level reconstruction loss, single-view geometric loss, multi-view consistency loss, and semantic feature distillation loss, the following new loss functions are introduced:

**Grid Smoothness Loss** ($L_{smooth}$)   Ensures smooth variation of the asymmetric grid tone mapper, aligning with CRF properties:

$$L_{smooth} = \sum_{i=1}^{N} \sum_{e \in [a,b]} g_i''(e)^2$$

**Unit Exposure Loss** ($L_u$)   Helps ensure the reconstructed HDR radiance field aligns with standard values, especially during HDR quality evaluation:

$$L_u = \|g(0) - C_0\|_2^2$$

The final total loss function is the weighted sum of all individual losses.

## 4  EXPERIMENTS

This section validates the proposed method and compares it quantitatively and qualitatively with existing approaches. We focus on large-scale autonomous driving scenarios, evaluating geometry reconstruction, novel view synthesis, and high dynamic range (HDR) scene reconstruction capability.

### 4.1  EXPERIMENTAL SETUP

#### 4.1.1  DATASETS

We primarily use the nuScenes dataset (Caesar et al., 2020) and the Waymo Open Dataset (Sun et al., 2020) to evaluate geometry reconstruction and view synthesis under complex, dynamic urban scenes.

**nuScenes.**    nuScenes is a multimodal dataset designed for autonomous driving research. It integrates six sensor modalities: six high-resolution cameras with 360° coverage, one 32-beam LiDAR, five radars with full-surround coverage, and high-precision GPS and IMU. It contains 1,000 scenes (700 for training, 150 for validation, and 150 for testing). Each scene spans 20 seconds of continuous recording, totaling over 1.4M frames of multimodal data. Data were collected in Singapore and Boston across diverse environments including urban streets, highways, and residential areas.

**Waymo Open Dataset.**    The Waymo Open Dataset is a large-scale resource offering high-quality multimodal data (point clouds, images, and annotations) for tasks such as detection, tracking, and semantic segmentation. Its sensor suite includes five LiDARs providing 360° coverage and five high-resolution cameras (front, rear, left, right, and top). The dataset comprises over 1,000 scenes (about 20 TB), spanning urban, suburban, and highway conditions, with large scale, precise annotations, and high scene diversity.

**HDRNeRF Dataset.**    To assess HDR reconstruction, we adopt the dataset released with HDR-NeRF, which includes 8 synthetic Blender-rendered scenes and 4 real captured scenes. Each scene provides 35 viewpoints with 5 exposure levels (from $-4$EV to $+5$EV) of LDR images.

### 4.1.2 EVALUATION METRICS

**Geometry Reconstruction (nuScenes).**    We report Chamfer Distance (CD) over multiple categories: full scene, pedestrians, dynamic vehicles, and background.

**Rendering Quality (Waymo).**    We evaluate novel view synthesis using PSNR, SSIM, and LPIPS, also broken down into full scene, pedestrian, and dynamic vehicle components where applicable.

**HDR-Specific Metrics (HDRNeRF).**    We report the HDR-VDP Q-score, perceptually uniform PSNR (PUPSNR) and SSIM (PUSSIM), in addition to conventional LDR metrics (PSNR, SSIM, LPIPS).

## 4.2 EXPERIMENTAL ANALYSIS

### 4.2.1 QUANTITATIVE ANALYSIS

**Ablation Studies.**    We conduct detailed ablations to verify the contribution of each component. In particular, the coarse-to-fine strategy, temporal scale scaling, asymmetric grid, and unit-exposure loss each improve HDR reconstruction quality and convergence speed. Results are shown in Table 1.

Table 1: Ablation study on HDR reconstruction quality. $\uparrow$ / $\downarrow$ indicate higher / lower is better.

| Setting | PSNR $\uparrow$ | SSIM $\uparrow$ | LPIPS $\downarrow$ |
|---|---|---|---|
| Baseline (3DGS) | 23.74 | 0.637 | 0.408 |
| + Unit Exposure Loss | 24.12 | 0.625 | 0.417 |
| + Coarse-to-Fine Strategy | 25.52 | 0.631 | 0.389 |
| + Temporal Scale Scaling | 25.38 | 0.785 | 0.384 |
| + Asymmetric Grid | 26.49 | 0.729 | 0.313 |
| Full Method | 27.32 | 0.810 | 0.212 |

**Geometry Reconstruction (nuScenes).**    As shown in Table 2, our method achieves lower Chamfer Distance across full scene, pedestrian, dynamic vehicle, and background categories compared to **Neurad** (Tonderski et al., 2024), demonstrating that geometry priors and multi-view consistency constraints effectively enhance reconstruction in complex dynamic urban environments.

**Rendering Quality (Waymo).**    On the Waymo Open Dataset, our approach achieves higher PSNR and SSIM than **OmniRe** (Chen et al., 2024), especially under multi-view inconsistencies caused by camera ISP variations (Table 3).

Table 2: Chamfer Distance (CD, lower is better) comparison on nuScenes.

| Scene | Method | Full | Pedestrian | Dynamic Vehicles | Background |
|-------|--------|------|-----------|-----------------|-----------|
| nuscenes-164 | Neurad (Tonderski et al., 2024) | 0.728 | 2.034 | 15.392 | 0.731 |
| | Ours | 0.667 | 1.926 | 12.375 | 0.673 |
| nuscenes-209 | Neurad | 1.036 | N/A | 6.843 | 1.018 |
| | Ours | 0.963 | N/A | 5.753 | 0.960 |
| nuscenes-359 | Neurad | 0.972 | 10.012 | 15.767 | 0.964 |
| | Ours | 0.953 | 9.136 | 12.439 | 0.947 |
| nuscenes-916 | Neurad | 1.634 | 8.628 | 8.093 | 1.636 |
| | Ours | 1.540 | 6.656 | 7.546 | 1.555 |

Table 3: Novel view rendering quality on Waymo Open Dataset.

| Scene | Method | Full PSNR | Full SSIM | Moving PSNR | Moving SSIM |
|-------|--------|-----------|-----------|-------------|-------------|
| waymo-023 | OmniRe (Chen et al., 2024) | 34.22 | 0.955 | 26.57 | 0.796 |
| | Ours | 35.84 | 0.965 | 30.84 | 0.913 |
| waymo-114 | OmniRe | 26.04 | 0.859 | 21.41 | 0.609 |
| | Ours | 29.81 | 0.928 | 25.48 | 0.777 |
| waymo-327 | OmniRe | 32.08 | 0.932 | 24.95 | 0.735 |
| | Ours | 33.49 | 0.949 | 29.26 | 0.876 |
| waymo-621 | OmniRe | 32.00 | 0.926 | 22.27 | 0.646 |
| | Ours | 34.65 | 0.962 | 29.68 | 0.910 |

**HDR Scene Reconstruction.**  On the HDRNeRF dataset, our method surpasses HDR-NeRF and HDR-Plenoxel across LDR and HDR metrics (PSNR, SSIM, LPIPS, HDR-VDP Q-score, PUPSNR, PUSSIM), and shows stronger robustness under the LDR-NE (sparse exposure) setting, validating the effectiveness of the radiance representation and asymmetric tone mapper (Table 4).

**Computational Efficiency.**  Although additional modules are introduced, the coarse-to-fine optimization strategy significantly reduces total training steps, making overall training time comparable to or shorter than baselines. During rendering, by pre-solving and caching material-related attributes, we still sustain real-time performance (over 100 FPS).

Table 4: HDR reconstruction performance comparison on HDRNeRF dataset. (Top) Conventional LDR metrics; (Bottom) HDR-specific perceptual metrics.

| Method | Dataset | PSNR ↑ | SSIM ↑ | LPIPS ↓ |
|--------|---------|--------|--------|---------|
| HDR-NeRF (Huang et al., 2022) | HDRNeRF | 36.72 | 0.951 | 0.044 |
| HDR-Plenoxel (Kim et al., 2022) | HDRNeRF | 36.42 | 0.932 | 0.043 |
| 3DGS (Kerbl et al., 2023) | HDRNeRF | 11.97 | 0.431 | 0.489 |
| Ours | HDRNeRF | 38.21 | 0.965 | 0.213 |

| Method | Dataset | Q-score ↑ | PUPSNR ↑ | PUSSIM ↑ |
|--------|---------|-----------|----------|----------|
| HDR-NeRF (Huang et al., 2022) | HDRNeRF | 6.56 | 20.33 | 0.527 |
| Ours | HDRNeRF | 9.71 | 22.57 | 0.735 |

### 4.2.2 QUALITATIVE ANALYSIS

**Geometry Prior Visualization.**  We visualize predicted depth and normal maps to illustrate how geometry priors guide correspondence matching and surface refinement, accelerating convergence, and improving reconstruction fidelity.

**View-Independent Consistency.**  Visual results show that bilateral filtering effectively suppresses inconsistent illumination (e.g., shadows) across input views, reducing overfitting artifacts in shadow

regions and improving stability in novel view synthesis. Furthermore, our method decouples exposure discrepancies caused by camera ISP, correcting color casts, and enforcing cross-view appearance consistency.

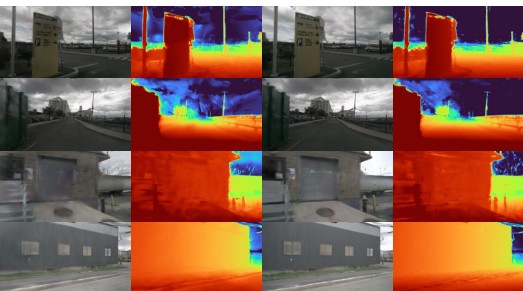

Baseline method          Our method

Figure 4: Comparison of the effects of view independence consistency

**Complex Materials and HDR Scenes.** Our approach faithfully reconstructs reflective objects, producing smooth and complete surfaces. It also converges faster than 3DGS-DR on specular intensity modeling, benefiting from incorporated high-dimensional semantic cues.

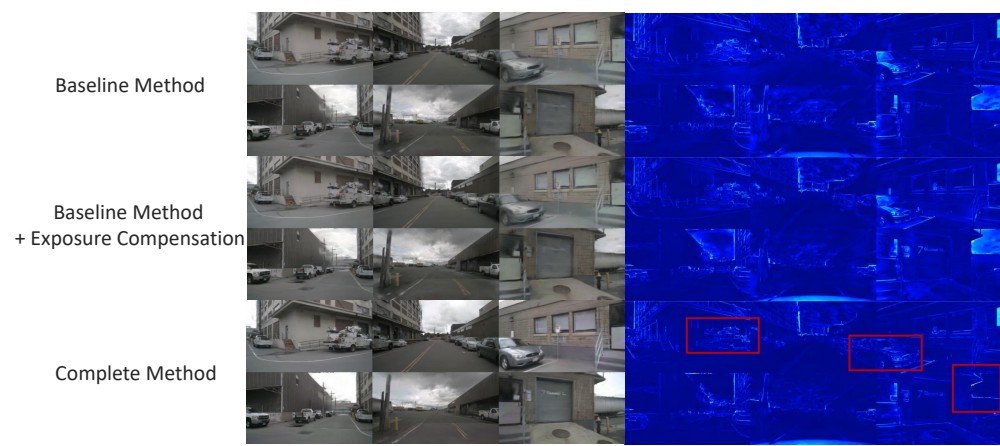

Figure 5: Comparison of effects between complex materials and HDR scenes

## 5 CONCLUSION

We introduced a unified neural rendering framework that improves sparse-view geometry, multi-view appearance consistency under ISP variation, complex material depiction with dynamic lighting, HDR radiance reconstruction, and training efficiency. Geometry and feature-level regularization stabilizes sparse reconstruction; a disentangled global-and-local appearance module suppresses artifacts; a semantics-guided implicit material dictionary with deferred rendering enhances material fidelity; radiance-based color plus a learnable asymmetric tone mapper boosts HDR accuracy; and a coarse-to-fine schedule speeds convergence while reducing overfitting. The resulting pipeline advances geometry, appearance coherence, material realism, HDR fidelity, and efficiency, forming a foundation for future dynamic and editable large-scale scene modeling.

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

## A  APPENDIX

### A.1  THE USE OF LARGE LANGUAGE MODELS (LLMS)

A large language model (OpenAI GPT-5) was used exclusively to suggest minor improvements in grammar, phrasing, and stylistic clarity after the full scientific content (ideas, methods, experiments, results, and conclusions) had been finalized by the authors. The tool was not used to generate text containing novel technical contributions. All model outputs were critically reviewed and selectively incorporated. The authors are solely responsible for the integrity and originality of the work, and the LLM is not listed as an author.

