# OpenReview forum: "High-Fidelity 3D Scene Representation via HDR-Integrated Multi-Constraint Neural Rendering"
_ICLR.cc/2026/Conference — Submitted to ICLR 2026_

### Official Review · Reviewer_Axr5 · 2025-10-19

**Soundness:** 3
**Presentation:** 3
**Contribution:** 2
**Rating:** 6
**Confidence:** 4

**Summary:**

This paper presents a unified framework for high-fidelity 3D scene representation based on 3D Gaussian Splatting (3DGS). The authors claim to address common failure modes of neural rendering, such as complex illumination, weak textures, and inconsistencies arising from camera ISP pipelines.

The paper claims four main contributions:

- A geometric reconstruction algorithm that introduces multi-view geometric consistency priors, using depth-guided virtual view generation to improve geometric accuracy in sparse and weak-textured scenes.
- A view synthesis algorithm that decouples camera ISP effects using bilateral filtering as a learnable proxy for ISP transformations. This component is claimed to be guided by large-model semantics and deferred rendering techniques to handle complex materials.
- The introduction of radiance as the color representation for Gaussian points, combined with a learnable asymmetric grid for tone mapping from HDR radiance to LDR colors.
- A coarse-to-fine optimization strategy for the tone mapper to improve stability and convergence.

The authors claim that their integrated approach achieves state-of-the-art results in geometry reconstruction, novel view synthesis, and HDR reconstruction, demonstrating robustness under challenging real-world conditions.

**Strengths:**

The paper tackles a significant and timely problem: improving the robustness and fidelity of 3D Gaussian Splatting for real-world captures. The ambition to create a unified framework that simultaneously addresses geometric inaccuracies, photometric inconsistencies (ISP), complex materials, and HDR content is commendable. The qualitative results presented in the paper appear compelling, showcasing clear improvements over baseline methods in challenging scenes. The idea of combining multiple constraints (geometric, photometric, semantic) into a single optimization process is a strong engineering effort and, if novel, could be a significant contribution to the field.

**Weaknesses:**

My primary concern lies in the overlap of its core ideas with prior and concurrent methods, and the performance achieved.

### Contribution 1 (Geometric Consistency):

The idea of using multi-view geometric priors to regularize neural representations is not new. The authors themselves cite foundational works in this area. More specifically, using photometric consistency over warped patches (rather than single pixels) to provide a stronger geometric signal has been explored. For instance, "Improving neural implicit surfaces geometry with patch warping" (CVPR 2022) [1] proposed optimizing an implicit geometry by warping patches between views and measuring their structural similarity (SSIM), which is conceptually very similar to the multi-view photometric consistency constraint mentioned here. The paper needs to better articulate how its specific implementation of "depth-guided virtual view generation and feature-level regularization" differs from and improves upon these established patch-based consistency methods and depth-based regularizations.

### Contribution 2 (ISP Decoupling & Semantic Guidance):

ISP Decoupling: The core idea of decoupling the physical scene radiance from the camera's tone mapping and ISP pipeline is a central theme in recent HDR neural rendering works. Several papers, including "HDR-GS" (NeurIPS 2024) [2], "GaussHDR" (CVPR 2025) [3], and the earlier "HDR-NeRF", have already implemented this by representing the underlying scene in a linear HDR space (radiance) and training a separate, learnable neural network (usually an MLP) to model the non-linear tone mapping function. The authors of this paper fail to cite or discuss these highly relevant works. Their specific choice of using bilateral filtering as an ISP proxy is an implementation detail, but the fundamental contribution of "decoupling ISP effects" has already been established in the context of HDR+GS.

Semantic Guidance: Leveraging semantic features from pre-trained models (like CLIP) as a consistency loss to regularize few-shot or challenging reconstructions is also a known technique. "Putting NeRF on a diet" (ICCV 2021) [4] introduced a semantic consistency loss to supervise a radiance field from arbitrary poses by matching features from a pre-trained visual encoder. This paper's claim of using "large-model semantics" appears to follow the same principle, but it is presented without comparison to such prior art.

### Contribution 3 & 4 (Radiance Representation & Coarse-to-Fine Optimization):

These two contributions are heavily overlapped with existing HDR+GS literature.

Radiance Representation: The fundamental shift from representing LDR color to HDR radiance in 3DGS was proposed by "HDR-GS" (NeurIPS 2024) [2] and "GaussHDR" (CVPR 2025) [3]. Both papers explicitly redefine the Gaussian attributes to represent radiance and employ a learnable tone mapper that takes radiance and exposure time as input, following the physical imaging model. The current paper presents this as a novel contribution without acknowledging these precedents.

Coarse-to-Fine Optimization: A two-stage optimization for the tone mapper is explicitly described in "GaussHDR" (CVPR 2025) [3]. That work first trains a global tone mapper and then enables a residual, local component in a second stage. This is functionally identical to the "coarse-to-fine" strategy claimed here.

### Questionable State-of-the-Art Claims:

The paper claims to achieve state-of-the-art results, but the provided quantitative metrics in Table 4 are unconvincing. For instance, on the HDRNeRF dataset, the proposed method achieves an LPIPS score of 0.213. This is substantially worse than the LPIPS score of its own reported baseline, HDR-NeRF (0.044), and also significantly worse than results from other recent, uncited works like "GaussHDR" (CVPR 2025) [3] (0.016 on a similar LDR reconstruction task). While the PSNR is higher, a much worse perceptual metric like LPIPS suggests that the rendered images may have noticeable artifacts, which contradicts the claims of achieving "high-fidelity" results. The paper should provide a more balanced discussion of its performance and temper its claims of superiority.

In summary, the paper's main weakness is that it frames a combination of existing ideas as a set of novel contributions. The lack of citation and comparison to several key, recently published papers in the specific domain of HDR Gaussian Splatting is a major flaw that undermines the paper's claims of originality.

**Questions:**

The primary concerns are about the novelty and the positioning of this work relative to existing literature. A response from the authors clarifying these points could significantly impact my assessment.

The core methodology for HDR reconstruction (Contribution 3 and 4) appears nearly identical to the approaches in "HDR-GS" (NeurIPS 2024) [2] and "GaussHDR" (CVPR 2025) [3]. Both use a radiance representation for Gaussians and a learnable tone mapper, with the latter also employing a coarse-to-fine training schedule. Could the authors please elaborate on the key technical differences between their HDR pipeline and these methods? Why were these highly relevant papers not cited or included in the experimental comparisons?

Regarding the ISP decoupling (Contribution 2), how does the proposed bilateral filtering approach fundamentally differ from training a learnable MLP-based tone mapper as done in "HDR-GS" (NeurIPS 2024) [2] and "GaussHDR" (CVPR 2025) [3]? Both techniques serve as learnable functions to map the rendered HDR radiance to the observed LDR images, effectively modeling per-view camera processing. What is the specific advantage of using a bilateral grid over a simpler MLP in this context?

The use of semantic features for consistency (part of Contribution 2) is reminiscent of the semantic loss in "Putting NeRF on a diet" (ICCV 2021) [4]. Could you clarify the novelty of your semantic guidance, especially in how it is applied to 3DGS and for what specific purpose (e.g., few-shot, material properties)?

For the geometric consistency priors (Contribution 1), methods like "Improving neural implicit surfaces geometry with patch warping" (CVPR 2022) [1] have already shown the benefit of using patch-level photometric consistency (e.g., SSIM loss on warped patches) to improve geometry. How does your proposed "depth-guided virtual view generation" provide a distinct advantage over such patch-warping techniques?

The quantitative results in Table 4 show that your method achieves a significantly worse LPIPS score (0.213) compared to the baseline HDR-NeRF (0.044) that you report, as well as other uncited works like "GaussHDR" (CVPR 2025) [3] (0.016). A higher LPIPS score often indicates poorer perceptual quality. Could you please explain this discrepancy? Why should your method be considered state-of-the-art when a key perceptual metric is weaker than prior art?

Given the significant overlap with multiple existing methods, could the authors re-frame their contribution? Is the main contribution the specific combination of these existing techniques into a single, robust system? If so, the paper should be rewritten to reflect this, and the experimental section should include ablation studies that demonstrate the synergistic benefits of this particular combination over using each component in isolation.

---

### Official Review · Reviewer_AAsq · 2025-10-31

**Soundness:** 2
**Presentation:** 2
**Contribution:** 2
**Rating:** 2
**Confidence:** 4

**Summary:**

This paper aims to obtain a unified HDR representation from multi-view LDR images through 3D Gaussian Splatting (3DGS) optimization. The method first enhances the geometric accuracy of 3DGS via multi-view consistency, and then introduces a physically-based rendering (PBR)–based material decomposition and tunemap optimization to recover HDR representations. The effectiveness of the proposed approach is validated on driving-scene datasets.

**Strengths:**

1. By introducing multi-view consistency loss and PBR-based rendering decomposition, the proposed method effectively helps recover clean and accurate geometric structures from multi-view images.
2. The paper provides visualization results in Figure 4,5 to support the effects of methods.

**Weaknesses:**

1. The proposed method largely integrates components from existing works: Section 3.2 corresponds to PGSR [1], the image embedding in Section 3.3 is also used in PGSR, and the PBR decomposition in Section 3.4 appears in GaRe [2].
2. The writing is confusing, particularly in the methodology section, where a substantial portion is devoted to describing prior methods, while the contributions of the proposed approach and the distinctions from previous works are insufficiently clarified.
3. The paper contains some overclaims: the title, abstract, and introduction suggest applicability to general scenes, but the experiments are only conducted on driving scenarios. The specificity of the scene should be clearly indicated in the title and abstract.
4. The experimental comparisons are insufficient, lacking state-of-the-art HDR reconstruction methods such as HDR-GS [3] and GaussHDR [4].

[1] PGSR: Planar-based Gaussian Splatting for Efficient and High-Fidelity Surface Reconstruction
[2] GaRe: Relightable 3D Gaussian Splatting for Outdoor Scenes from Unconstrained Photo Collections
[3] HDR-GS: Efficient High Dynamic Range Novel View Synthesis at 1000x Speed via Gaussian Splatting
[4] GaussHDR: High Dynamic Range Gaussian Splatting via Learning Unified 3D and 2D Local Tone Mapping

**Questions:**

The authors are encouraged to clarify the specific improvements of their approach compared to the referenced methods, particularly in the methodology section. In addition, it would be helpful to explicitly highlight the advantages of the proposed method over existing HDR 3DGS approaches.

---

### Official Review · Reviewer_EzT2 · 2025-11-01

**Soundness:** 1
**Presentation:** 1
**Contribution:** 1
**Rating:** 2
**Confidence:** 4

**Summary:**

This paper presents a unified neural rendering framework that integrates HDR modeling, multi-view geometric consistency, and semantic-guided material modeling into 3D Gaussian Splatting (3DGS). The proposed system enhances geometric accuracy in weak-textured regions, enforces view-independent consistency via bilateral filtering to decouple camera ISP effects, and introduces a learnable asymmetric tone mapper for HDR scene reconstruction.

Experiments on nuScenes, Waymo, and HDR-NeRF datasets demonstrate improved geometric fidelity, photometric accuracy, and HDR rendering quality compared with Neurad, OmniRe, and HDR-NeRF.

**Strengths:**

1. The combination of multi-view geometric priors, bilateral filtering for ISP decoupling, and learnable asymmetric tone mapping is relatively novel. It addresses practical limitations of 3DGS under complex illumination and sparse inputs.
2. Quantitative and qualitative improvements across multiple benchmarks (nuScenes, Waymo, HDRNeRF) are clear and consistent. Ablation studies show a meaningful contribution from each proposed module.

**Weaknesses:**

1. The writing of this paper indicates it is not a completed version: many concepts are not clearly introduced, making it hard for readers to follow. The method section is dense, with intertwined submodules and inconsistent notation. Figures lack sufficient detail to trace data flow. Some equations omit variable definitions.
2. The novelty of this paper is limited: The radiance-based representation and tone-mapping grid are not new; the main novelty lies in integrating HDR handling with geometric and semantic consistency. The paper should clarify this distinction instead of implying a new HDR formulation.
3. The main issue of this paper is that, although many HDR 3DGS works proposed in 2024–2025 already employ radiance-based Gaussian color and tone-mapping strategies, they are not included in comparisons. This omission significantly weakens the HDR contribution claim and prevents a fair assessment of novelty.
4. Despite claiming “>100 FPS” rendering, the paper provides no quantitative runtime or throughput comparison (e.g., FPS vs. 3DGS, HDR-GS, OmniRe). For a 3DGS-based method, omitting performance metrics undermines the claimed real-time practicality.

**Questions:**

I believe the current version of this paper is not suitable for inclusion in ICLR 2026 and recommend that the authors revise the paper and resubmit it to other conferences.

---

### Official Review · Reviewer_iTTh · 2025-11-02

**Soundness:** 2
**Presentation:** 2
**Contribution:** 2
**Rating:** 2
**Confidence:** 4

**Summary:**

The paper proposes a 3D Gaussian Splatting (3DGS) based pipeline targeting high-fidelity reconstruction under difficult illumination, weak texture, and cross-view ISP inconsistencies. The system aggregates a) multi-view geometric consistency with depth-guided virtual views plus single-/multi-view losses to stabilize sparse/low-texture regions; b) view-independent consistency module that assigns per-view bilateral-grid transforms (and global embeddings/uncertainty masks) to decouple ISP variation from the radiance field; c) semantics-guided deferred shading via an implicit material dictionary for reflectance/specular; and d) an HDR formulation that represents color as radiance with a learnable asymmetric tone-mapping grid trained in a coarse-to-fine schedule. Experiments on nuScenes, Waymo, and HDR-NeRF data report improved geometry and photometric qualities relative to selected baselines.

**Strengths:**

1. The proposed pipeline incorporates 3 practical improvements into the 3DGS pipeline, targeting sparse/weak-texture geometry, ISP-induced appearance drift, and HDR tone-mapping. This is a useful systems contribution. The asymmetric tone-mapping grid and coarse-to-fine schedule are reasonable design choices aligned with HDR radiance-field literature and shown good performance in their experiments.
2. If validated rigorously, as a "drop-in" 3DGS pipeline that is more robust to ISP/HDR and low texture situations could be a strong baseline for difficult real world data such as in autonomous-driving and robotics scenarios, where exposure inconsistencies and harsh lighting are common.

**Weaknesses:**

1. Novelty relative to prior HDR & appearance-decoupling work is unclear, many core ideas closely mirror existing literature and not mentioned/discussed extensively:
a) HDR formulation (radiance + exposure/CRF + tone-mapping) follows HDR-NeRF and HDR-Plenoxels closely; the `asymmetric grid` resembles prior grid-based tone mappers/HDR-GS variants. The authors should more carefully discuss what is new beyond adopting a grid and a two-phase schedule.
b) Appearance/ISP variation is long addressed by per-image appearance embeddings and transient terms (e.g. in NeRF-W and Block-NeRF). Here, replacing the latent with per-view bilateral grids echoes earlier "deep bilateral learning" pipelines such as hdrnet, which model local affine transforms in bilateral space. The authors should add more discussions and justify why a bilateral grid (vs. embeddings) is necessary and superior for 3DGS- such design choices are not validated in the paper.

2. Writing: In general, the related works section is superficial and does not included many important field of works, such as those mentioned above. The section would require a complete rewrite.

3. Limited and uneven baselines in main experiments, incomplete ablation studies.
a) The main experiments only cover a very limited selection of baselines and are missing many directly comparable works. For example, on Waymo, comparisons are only to OmniRE; on nuScenes, only to NeuRAD. Competing 3DGS variants for urban scenes (e.g. SplatAD) and appearance-robust NeRF/Block-NeRF-style systems are absent. Likewise, for HDR, numbers versus HDR-GS variants are missing- even though these are directly comparable 3DGS HDR methods.
b) Most of the experiments, if not all, are missing important experiment settings such as data splits, evaluation protocols, baseline details, etc.
c) While some HDR ablation is shown, there is no clean ablation isolating many other design choices, e.g. bilateral grid vs. simple per-image appearance embedding, the contribution of uncertainty masking, view consistency, and many more. This makes it hard to attribute improvements.

**Questions:**

Please check the weaknesses sections very carefully for my major concerns.

---

### Meta-Review · Area_Chair_LDe5 · 2025-12-25

**Summary:**

The paper initially received mostly negative reviews, with scores of 2, 2, 2, and 6.

Reviewers identified several weaknesses in the paper, such as unclear novelty, insufficient experiments, inadequate baselines, limited ablation studies, and poor presentation quality. The authors did not provide a rebuttal to these concerns.

The area chair agrees with the reviewers' evaluation and recommends rejecting the paper.

**Reviewer Concerns:**

The authors did not respond to the reviewers' concerns.

**Reviewer Scores:**

The area chair expects the reviewers to maintain their initial scores.

---

### Decision · Program_Chairs · 2026-01-26

Reject